# A Transdisciplinary Approach to Recovering Natural and Cultural Landscape and Place Identification: A Case Study of Can Moritz Spring (Rubí, Spain)

**DOI:** 10.3390/ijerph18041709

**Published:** 2021-02-10

**Authors:** Marina Cervera, Simon Bell, Francesc Muñoz, Himansu S. Mishra, Lora E. Fleming, James Grellier, Glòria Carrasco-Turigas, Mark J. Nieuwenhuijsen, Cristina Vert, Mireia Gascon

**Affiliations:** 1Departament d’Urbanisme i Ordenació del Territori (DUOT), Universitat Politècnica de Catalunya- Barcelona Tech (UPC), 08028 Barcelona, Spain; mcerveraalonsodemedina@gmail.com; 2Institute of Agricultural and Environmental Sciences, Estonian University of Life Sciences, 51014 Tartu, Estonia; Simon.Bell@emu.ee (S.B.); HimansuSekhar.Mishra@emu.ee (H.S.M.); 3Edinburgh School of Architecture and Landscape Architecture, University of Edinburgh, Edinburgh EH1 1JZ, UK; 4Departament de Geografia, Universitat Autònoma de Barcelona, 08193 Barcelona, Spain; franc.munoz@uab.es; 5European Centre for Environment and Human Health, University of Exeter Medical School, Truro TR1 3HD, UK; L.E.Fleming@exeter.ac.uk (L.E.F.); J.Grellier@exeter.ac.uk (J.G.); 6Institute of Psychology, Jagiellonian University, 31-007 Krakow, Poland; 7ISGlobal, 08193 Barcelona, Spain; gloria.carrasco@isglobal.org (G.C.-T.).; mark.nieuwenhuijsen@isglobal.org (M.J.N.); cristina.vert@isglobal.org (C.V.); 8Department of Experimental and Health Sciences, Universitat Pompeu Fabra (UPF), 08193 Barcelona, Spain; 9CIBER Epidemiología y Salud Pública (CIBERESP), 28029 Madrid, Spain

**Keywords:** natural environments, landscape architecture, heritage, social participation, community engagement, health and wellbeing

## Abstract

The perception of the quality of green and blue spaces can be key in the relationship between a community and its local landscape (i.e., place identification). The lack of transdisciplinary training and social-specific education of landscape architects regarding the complexity of landscape as a participative cultural artefact limits reaching the general population. Bridging this gap of landscape and place identification and evaluation by a local community was the main objective of the present case study conducted at an abandoned spring and seasonal stream area in Rubí (Spain). The “Steinitz method” of landscape evaluation was used as a participatory method to activate community members to learn about and express their visual preferences regarding this neglected landscape. Bottom-up interventions applying an “urban acupuncture” approach in the area identified as the least attractive by the residents were co-designed and combined with a top-down restoration of a nearby, existing but derelict and hidden, spring. In addition, before and after planning and implementing the intervention, we conducted surveys about the community perception, sense of belonging and use of the space. We observed that the lack of awareness of the inhabitants about this spring was an obstacle preventing the community from embracing the potential for health and wellbeing presented by the spring and adjacent landscape. Following the work, the landscape saw increasing use, and the historic spring was brought back to life as a resource to help people to improve their health and wellbeing.

## 1. Introduction

Extensive literature from the last decade has provided increasing evidence of the health benefits of being exposed to natural environments, including nature in urban areas. Research results are remarkably consistent in demonstrating an association between green spaces (trees, grass, forests and parks) and blue spaces (rivers, lakes, springs and the coast) and better mental health and wellbeing [1,2,3,4]. Two recent systematic reviews showed a reduction in mortality associated with increasing exposure to green spaces close to residential areas [2,5]. Stress reduction, the promotion of physical activity and an increase in social interactions are some of the suggested mechanisms behind such health and wellbeing benefits. Such spaces also attenuate the adverse health effects of noise, air pollution and increased temperatures [6,7], which are mainly an issue in urban areas that are developing and densifying [8].

While existing studies show that quantity factors (e.g., the proportion of residential green space, distance from the coast or time spent in these spaces) are significant factors affecting health impacts, a limited number of studies have also indicated that the actual and perceived quality of green and blue spaces is essential for obtaining a range of positive health and wellbeing outcomes; this includes the perception of signs of pollution (e.g., foam on water), signs of a lack of stewardship (e.g., litter and vandalism), algal blooms, or low levels of biodiversity and wildlife [9,10,11,12,13]. In this sense, the concept “therapeutic landscape” was coined in 1992 to explore why specific environments seem to contribute to a healing sense of place [14,15]. Besides, access to nature is an even more critical factor [16,17,18,19]. Bearing this in mind and considering that a recent study observed that there is an exponential relationship between the residential distance to green and blue spaces and visits to these sites [20], it is crucial to conduct interventions co-created with affected individuals and communities in order to ensure the availability of good-quality nearby green and blue spaces. Moreover, such interventions can reduce the inequality of access to such environments and, in turn, reduce health and environmental inequalities, as we have shown in recent intervention studies [17,21].

Since the European Landscape Convention (ELC) [22] was adopted in 2000, a renewed concept of landscape spread throughout the continent. The ELC defines landscape as “an area, as perceived by people, whose character is the result of the action and interaction of natural and/or human factors”. This notion has been widely accepted by scientific literature [23] and has boosted the Convention as the most widely adopted (now ratified by 40 out of 47 CoE members). It is also a globally influential approach to the subject (through its amendment opening the text to countries outside Europe [24]). Through its Article 5, the ELC recognizes landscapes to be an essential component of people’s surroundings and a foundation of their identity because they are an expression of the diversity and shared cultural and natural heritage of a specific community [25]. All the signatories to the Convention commit, in Article 5c, “to establish procedures for the participation of the general public, local and regional authorities, and other parties with interest in the definition and implementation of the landscape policies mentioned in paragraph b above”. This section secures the basis for collaborative design and the community’s empowerment to reflect and intervene in their landscapes. The notion of the Right to Landscape [26] evolved from the ELC principles and the shared social responsibilities to the environment, emphasising the importance of governance beyond the definition of landscape, extending the specific awareness-raising measures integrated into Article 6 of the ELC. 

The importance of landscape perception and public participation is thus only fostered in European-level policy, whilst a broader range of objectives on health and wellbeing and the importance of life on land are recognized in the 2030 Sustainable Development Agenda, specifically through sustainable development goals 3 and 15 [27]. Even though scientific literature has identified the strong links between landscape and health [8,14,19], the importance of landscape regarding community health and community building seems to have been disregarded in international recommendations and sustainability agendas, which explains the focus on the ELC as the main reference for this research. Understanding that not all communities are cohesive and not all individuals have the same levels of awareness towards their landscapes, the ELC promotes a new culture of participatory and collaborative landscape analysis, management, and design, focusing on ordinary landscapes as opposed to previous concerns over highly valued or protected landscapes.

The recognition of the importance of the ordinary urban landscape can be traced back to the 1990s [28] and is still integrated into environmentalist mainstream thinking, entrenching equity of access to the landscape as a right. The notion of preserving natural landscape as heritage can be traced back to 1872, when Yellowstone was protected as the first U.S. national park, becoming an area of exceptional natural landscape protection and a touchstone of the emerging environmental culture. The time needed for the academic contribution on ordinary or every day landscapes to reach public acceptance and the integration of ordinary landscape as a collective right is tackled in the ELC text through an emphasis on education [29]. The ELC encourages interdisciplinary training programmes in politics, nature conservation and landscape management aimed at professionals in the private and public sectors, and specific training in schools and universities. However, the lack of awareness towards landscape complexity as a cultural artefact among the general public is an obstacle.

While the landscape and the ELC are rightly considered important, discussing the local landscape or environment with communities reveals that landscape as a concept is better replaced with the notion of “place”. Depending on the authors, places are a combination of the physical environment, the activities people do there, and the perceptions they hold [30]. When researchers—trained as landscape architects—try to interview the residents of a particular locality about their landscape, they often find it frustrating that the interviewees’ responses wander away from the specific physical environment to talk about living there, jobs, services and other things (which are indivisible from the place, its identity and the attachment people have for it). The limited social- and participation-specific education in landscape architecture curricula [31,32,33] and lack of transdisciplinary training is challenged by the collaborative and multidisciplinary case-study approach embraced by authors from different backgrounds. Thus, while it is important to consider landscape as a subject field, practically speaking, it can be a closed book to local people unless they can be given the means to understand it, such as through public participation and co-design processes. The tripartite model of Scannel and Gifford is a well-known approach for understanding place and has been widely used in local participation processes [34,35].

Bridging this gap of landscape and place identification and evaluation by local residents was the main objective of the present case study conducted in the municipality of Rubí (Barcelona, Spain). The case study was the result of interdisciplinary and trans-sectoral work involving environmental epidemiologists and landscape architects within the E.U. Horizon 2020-funded BlueHealth project, together with landscape architect practitioners and academics in the area of Barcelona, students of the Master in Landscape Intervention and Heritage Management (LIHM, at the Universitat Autònoma de Barcelona—UAB) course, a local non-governmental organisation (NGO) (Rubí d’Arrel) with expertise in nature and heritage protection and recovery, the local public administration (City Council of Rubí), and, most importantly, local citizens. The specific objectives of the case study, as a joint research and practice project, were (i) to renovate the spring of Can Moritz (Rubí) and its surroundings; (ii) to co-design the renovation intervention by engaging the residents in a comprehensive participatory process integrating the analysis, design and partial construction of the particular place; (iii) to empower and inform citizens regarding their responsibilities and rights over their everyday landscapes; and iv) to evaluate the use and perception of the place before and after the intervention.

## 2. Materials and Methods

The BlueHealth project (https://bluehealth2020.eu), which ran from January 2016 to December 2020, aimed to understand the relationships between exposure to blue space and health and wellbeing, to map and quantify the public health and wellbeing impacts of changes to natural and artificial blue spaces and associated urban infrastructure in Europe, and to provide evidence-based information to policymakers on how to maximise the health benefits associated with interventions in and around aquatic environments [36]. Following the latter objective, eleven small-scale experimental interventions were conducted and evaluated in different European regions (https://bluehealth2020.eu/research/). One of these interventions took place in the municipality of Rubí (Barcelona, Spain), an industrial city of 77,500 inhabitants located 20 km from Barcelona, and with practically 50% of its land classified as nondevelopable (i.e., protected from being built on). Rubí is characterised by a combination of a dense city centre, industrial parks, and a significant expanse of low-density residential areas (dispersed) mixed with nondevelopable land. The BlueHealth intervention aimed to recover and renovate, in collaboration with residents, an abandoned historical site, the spring of Can Moritz, together with some adjacent derelict public land along a small seasonal stream fed by the spring, in a valley next to one of the communities in this low-density residential area (the valley of “Les Martines”), located a 20 min drive from the city centre (Figure 1).

### 2.1. The Study Site: The History of the Can Moritz Spring and Its Surroundings

According to local media and records from the City Council of Rubí, in the mid-19th century, Louis Moritz, the founder of the well-known Moritz Brewery in Barcelona, bought the property of “Can Matarí” (“Can” in Catalan means “the house of”) in the “Les Martines” valley, Rubí, as a summer house. The “Vallès” region, where the municipality of Rubí is located, and specifically, the lush and shady valley of “Les Martines”, was targeted at that time as a “nature destination” offering better health conditions during the hot summer periods for the upper-middle-class citizens of Barcelona. The urban living conditions in Barcelona’s dense medieval central city, walled until 1854, rendered basic hygiene features (fresh water, clean air and contact with nature) a privilege to be sought in the neighbouring countryside. The “Can Matarí” farmhouse was thus part of a cultural landscape that featured bourgeois summer houses on the “Vallès” hillsides and connected the main roads to the land around these houses, traditionally comprising the main house, the surrounding gardens, its productive lands (linked to the original farmhouse) and private springs. These springs offered fresh water and an opportunity to sit within a humid enclosure to feel the cooling effect.

After the death of Louis Moritz, in 1922, the family built a modernist-style house next to the existing farmhouse (“Can Matarí”) and designed a similarly styled enclosure around the natural water spring initially used by the landowners for their water supply, which is currently named after Moritz (Can Moritz). This spring comprised an oval or “bathtub”-shaped brick-lined tank set into the ground, at the base of which the spring water was fed into channels. Around the edges were located several brick benches resembling sofas upon which people could recline, relax and feel cooler in the shady and moist atmosphere.

Around the mid-1950s, the family sold the property, and the land was parcelled out, which resulted in the house of Can Moritz remaining in private ownership, currently catalogued as the archaeological heritage of Rubí. By contrast, the surrounding plots were sold and developed as a low-density garden city or suburban housing. The spring belonging to Can Moritz ended up in public land, as it was located next to a seasonal stream. The stream basin was integrated into the state hydrological water protection area and thus remained cut off from the summer house system. Scarcely noticed by local people, it was soon abandoned and gradually infilled and overgrown, that it was utterly lost. In December 2014, after a massive heavy storm, the spring and its modernist recreational structure were rediscovered by residents. In 2016 a local NGO, Rubí d’Arrel, instigated the first restoration intervention with volunteers by removing the vegetation, and cleaning the spring and the surroundings, unveiling the 1922 design for recreational uses around Can Moritz’s spring (Figure 2). Simultaneously, the area across the stream was identified as needing some environmental improvements—in part, so that a footbridge could be laid to improve access to the spring (which was at the base of a steep slope), and to revitalise and restore the landscape.

### 2.2. The Approach

In 2016, a collaboration between the BlueHealth project, the local NGO Rubí d’Arrel, professors and students of the Master programme in Landscape Intervention and Heritage Management (LIHM), and the City Council was established to conduct an extensive intervention to renovate the spring and its surroundings while involving the neighbours of the Can Moritz spring. While the City Council assumed the costs of renovating the spring itself as a heritage architectural structure (the most expensive part of the intervention, ≈ EUR 80,000), the BlueHealth project spent a modest amount (EUR 10,000) to conduct what is known as an “urban acupuncture” (i.e., a small-scale intervention to transform the broader urban context around the spring and its nearby stream). An integrative project for the whole study area would never be practical in budgetary terms. However, urban acupuncture builds on the progressive institutionalization of grassroots initiatives to shape the city, as tactical urbanism means relational processes in space testing [37,38].

The full project took place between 2016 and 2020 (Figure 3). In 2017, we engaged and recruited neighbours for our preintervention evaluation to obtain information on the use and perception of the study site, and conducted workshops that enabled the students of the Master in LIHM programme to design the intervention, which was conducted between 2018 and 2019, together with the restoration of the spring. In 2020, we conducted the postintervention evaluation (see Figure 3). We carried out a multimethod study comprising a questionnaire survey and two public participation landscape/place evaluation workshops. The survey was carried out pre- and postintervention implementation.

### 2.3. Design of the Intervention

The students of the Master in LIHM programme conducted two workshops. In the first, the focus was on a landscape analysis of the site of Can Matarí (“Les Martines”), including the valley and hillsides along the Can Matarí stream where the house and the spring are situated. The students were mentored to learn about the site’s topographical, natural and historical characteristics, interweaving the objectives and subjective framework of cultural landscapes [39,40]. The analysis followed the classic method of the “layer cake model” for the site developed and published by Ian McHarg during the 1980s [41], building upon foundations laid by Sauer, where the landscape is understood as being built up in layers from the geology, soils, hydrology ecology and cultural processes. This method contributed to building ecologically based land use planning, which should be regarded as a fundamental theory for guiding future developments on a specific site. This workshop was held with the collaboration of the City Council. 

The second workshop (Figure 4) consisted of a participatory workshop where students were commissioned to trigger landscape/place awareness among the local community in the “Les Martines” valley. In this case, we adapted the visual preference methodology of Steinitz [42,43], originally meant for evaluating natural environments, to evaluate urban environments. This methodology was finetuned to map local inhabitants’ visual preferences and thus detect, through a visual preference heat map, where landscape improvement and, thus, design intervention were seen as being most necessary. Steinitz’s overall aim of reconciling the potentially conflicting aspects of places that are both highly visually preferred and identified as having high ecological integrity was adapted to foster landscape awareness-raising and achieve a sustainable landscape through governance rather than planning. The elaboration of visual preference mapping (VPM) was the main focus of the second workshop, which was challenging, given the characteristics of the community and its local landscape. The character of the urban settlement structure along the long street running parallel to the stream was the primary condition determining the community’s use and behaviour around the “Les Martines” valley. In addition, the low-density urban sprawl carried low numbers of residents in direct relation to the spring and the nearby green spaces, and determined their dispersion. The workshop was thus targeted at a widely spread community with little interaction with each other, minimal knowledge of the immediate local environment as a result of most people’s commuter lifestyles, and little identification with their local landscape. 

We summarize the steps that were taken between 2017 and 2018:

(1) The first contact with the neighbours was the joint analysis of their built environment through the visual mapping workshop activities. After several weeks of research on the social, environmental and landscape conditions, the master students prepared a face-to-face session. This focused on the evaluation of a large sample of pictures representative of the nearby urban landscapes. This assessment was based on both the Steinitz method (see above) and the methodological guidelines for landscape studies used previously in the Valencia Orchard project to foster green infrastructure planning at different scale levels [44,45]. The degree of preference of the community in the “Les Martines” valley for those images representing the ordinary landscapes was compared with the expert evaluation of the same scenes according to a hypothesis suggested by the team of teachers and students formulated according to landscape perception principles developed since the 1970s by Kaplan and [46,47]. Recreational activities were offered along with the workshop with semistructured interviews with some neighbours, which was helpful for the later participatory stages.

(2) Once the first session results had been compiled and analysed by applying a linear regression model between the grading of each picture as evaluated by the community and the expert team according to Kaplan’s hypothesis, the resulting visual preference was mapped. The heat map of visual preference (low to high) as rated by the neighbours was analysed and used for feedback and strategic work. Throughout the different sessions, a draft global landscape management and intervention project were used, with the final aim of producing and invigorating the public space identified as the most preferred location next to the old rediscovered spring of Can Moritz. An overall masterplan was co-designed, following the City Council administration’s inputs, with academics serving as advisers, local residents, Rubí d’Arrel (NGO), and other associations. Finally, a set of actions and interventions applying the vocabulary of the urban acupuncture and tactical urbanism [48] used by grassroots movements was articulated into an action plan that would enhance the design vision while triggering the direct participation of the community. The overall aim was to design a landscape infrastructure that connected the existing fragmented urban sites and heritage elements to the natural features (blue–green infrastructure), adding value to the low-density residential area, revealing previously hidden cultural landscapes and improving the landscape image.

(3) All the stakeholders organized the final and main participatory session to produce the public space as agreed at the masterplan level. At this stage, the urban acupuncture and tactical urbanism actions were put into place, by a varied set of synchronous and parallel activities to be realised (hands on) by the residents with input from the volunteers of Rubí d’Arrel and the overall guidance of the masterplan coordinators. The actions were intentionally designed to take place during the same weekend, which required a lot of coordination and time and people management. This part of the project was funded by the BlueHealth project and included inputs by two landscape architects, who coordinated the overall urban acupuncture part of the research and held the research budget to pay for the construction and planting.

### 2.4. The Pre-/Postintervention Survey

We conducted a questionnaire survey in Catalan and Spanish, based on the BlueHealth Community Level Survey (BCLS) [36], and included questions on personal characteristics (gender, age, education level, work status and general health), the frequency of visits to natural environments (in the last 12 months), whether they knew the Can Moritz spring and if they had visited it in the last six months and in the last four weeks, their opinions about the quality of the site, the activity conducted there, time spent, etc. The same questions were asked before and after the intervention. 

We recruited participants (≥16 years of age) through different strategies: (i) online, using a Twitter account specifically created for the project (@FontCanMoritz); (ii) advertising the study in local media; (iii) leaflets handed out during the local festival in Rubí (many people are in the street at that time); and (iv) leaflets distributed to the mailboxes of the residents closest to the Can Moritz spring. The participants could answer the survey online or fill in a paper version and send it back via regular mail. 

## 3. Results

### 3.1. Visual Preference Mapping (VPM) and the Intervention

In total, 17 local people participated in the workshop for the VPM. The results can be seen in Figure 5. In summary, the VPM revealed potential hotspots of conflict and neglect (in red) while revealing the potential of the green–blue corridor’s line along the stream (in green) as being the highest visually preferred feature. 

In addition, some of the VPM revealed the influence of senses beyond the visual in the mapping results, uncovering the potential of the stream in which the spring is located. For instance, the main road’s reddish colour may be considered a contradictory finding, as this high-level, winding road offers the most scenic views over the valley but was poorly rated by the inhabitants. This might be related to its character as busy, noisy and dangerous for pedestrians since it features no walkway on the riverside, where the slope is abrupt. 

These findings led the masterplan development to establish a continuous path along the river, integrating the Can Moritz spring as the main recreation site in the valley. The main challenge was to tackle the sizeable negative hotspot around a car parking area, which the community perceived as the least preferred (most disliked). The pictures and the semistructured interviews identified illegal activities and night-time vandalism taking place there, leading to the accumulation of litter and pollution of the stream. The lack of care and stewardship in the flat area around and within the parking area was also perceived as an obstacle to the desired path route following the stream basin. 

The masterplan formalized a series of hands-on actions to define the desired line for the path along the stream as passing through the most fragile section across the parking area and its immediate surroundings (Figure 6). The purpose of these small-scale tactical interventions was to claim back the area perceived as the least attractive by mobilizing the site potential and future care by the local community. The collaborative activity included painting blue circles on the car park surface; arranging cut logs into a pattern, also painted blue so as to produce a visual connection across the site; planting trees and bushes; and adding furniture (Figure 7).

### 3.2. The Survey 

In total, 86 inhabitants of Rubí (not necessarily living near the study site) participated in the survey before the intervention. After the intervention, we obtained answers from 43 of these participants. Their characteristics are shown in Table 1. In summary, before the intervention, the participants were between 21 and 77 years of age (average, 46.8), 45.4% were female, and 46.7% had higher education (university degrees). Ninety-four percent reported having access to a garden at home (either a community or private garden). Sixty-four percent knew of the Can Moritz spring, but 34.6% had not visited it in the last six months, and among those who had, only 14 (38.9%) had visited it in the last four weeks. 

When comparing the total population (N = 86) with the population who also answered the postintervention survey (N = 43), we observed that the sample characteristics were similar except for (i) the percentages of people working at the time of the survey (66.3% vs. 79.1%, respectively); (ii) involvement in local NGOs, organizations or entities (33.7% vs. 51.2%, respectively); and dog ownership (48.8% vs. 37.2%, respectively). Other major differences in the postintervention group were regarding the place where people reported they worked (in Rubí or not) and that more people knew of the Can Moritz spring (increased from 60.5% to 83.7%). The postintervention participants who had visited the site in the last four weeks reported spending more time there (increasing from 15.8 min on average before the intervention to 32.5 min after the intervention) (Table 1).

Overall, the quality of the spring and its surroundings was mostly rated as “bad” or “very bad” before the intervention (≈40%), whereas after the intervention, more than 50% of the participants rated the site as of “good” or “excellent” quality (Table 1). When asking for detailed information on how they felt about their visit (in the last four weeks) to the Can Moritz spring, we observed that the levels of satisfaction (rated as “totally agree”) substantially increased from less than 20% to more than 60% (Table 2). Additionally, after the intervention, the participants more frequently felt part of nature, very few reported feeling unsafe, and the presence of rubbish and litter was less of an issue. Despite the improvements, however, the participants felt that the facilities (parking, roads, toilets, drinking water points and barbecue sites) could still be improved (Table 2).

We also asked the participants to provide their opinions (open answers) regarding the site before and after the intervention. Before the intervention, many comments reflected and complemented what was captured by the VPM. Some of the comments include: *“[…] I would very much like to see its restoration and its environment as well as its conservation, I have practically lived here all my life and, since I was a child, I have visited that spring, always flooded, and over time covered by vegetation and even, so it seems, a place of special charm. I usually walk around the area with my dog and pass by the spring. I have always been curious about what the spring would be like, I am a great lover of nature and these spaces […]”* (male, 47 years).*“It would be very interesting to take advantage of this space since it is part of our historical memory”* (male, 51 years).*“It needs to be opened to the public and to commemorate its past and history”* (female, 55 years).*“At the moment, access is a bit difficult, and the spring is in poor condition. I look forward to your rehabilitation”* (female, 74 years).*“Everything is destroyed, there is a platform with a hole, and I do not dare to walk on it because of the danger of it breaking”* (male, 74 years).*“The location of the spring should be indicated, and young people should be monitored so that it is not destroyed”* (male, 77 years).

After the intervention, the comments were much more positive, such as: *“We are really happy with the spring of Can Moritz to recover a piece of our past. Thanks to people like you, we have a more beautiful and natural place […]”* (male, 51 years).*“I totally agree with the project and the initiative to promote this space, I am concerned about the dumping of garbage in the surroundings that influence this space, because although there are containers for these purposes, people’s awareness is low! Plastics, cans, papers, which do not sit well with this environment”* (male, 47 years).*“The spring of Can Moritz, now recovered, promotes encounters with other people, in a natural space”* (male, 79 years).*“A pleasant space, a very well achieved heritage recovery”* (female, 63 years).

## 4. Discussion

The Can Moritz case study showed a relationship between landscape perception, the use of shared spaces and community wellbeing; the perceived low quality of the natural environment along the stream in which the Can Moritz spring is located was limiting the use of this common space. The Steinitz methodology helped to identify the most and least preferred areas on a visual map to inform the strategy for intervening on the site and improving the neighbours’ impressions of their open spaces. We also demonstrated the importance of collaborative work among different actors and the combination of top-down (the municipality taking responsibility for the spring) and bottom-up (the co-design and the application of “urban acupuncture” and tactical urbanism in the rest of the project area) strategies.

The present case study was particularly challenging regarding the ELC definition of landscape because we were targeting an ordinary, rather bland urban sprawl area on the outskirts of Rubí old town, where a low-density community inhabited a range of isolated housing in an undervalued hilly valley. The case study demonstrates how “urbanalization” (a spatial and cultural process characterized by a territorial specialization that accompanies trends in the thematization and simplification of the urban landscape [49]) led to the degradation of the landscape in the valley, where natural landscapes highly praised in the 19th century, leading to building a new summer community in the surroundings, declined over time into an unnoticed and undervalued environment. 

The Steinitz methodology had previously been applied for similar purposes in several workshops within Barcelona’s city, within the Master in LIHM programme directed by Francesc Muñoz (e.g., the Pere IV Workshop in 2014 and Playful Gràcia District in 2015). These previous studies had proven the effectiveness of using the methodology in urban environments, where the combined activities of university staff and students energized interaction and co-creation with residents and other stakeholders. 

In the present case study, framed in a less urbanized area and with a strong presence of natural environment, the location and identification of the least preferred landscapes signalled the community’s concerns, including signs of water pollution and algal blooms, which proved to be minor problems compared to the signs of a lack of stewardship and the presence of ongoing vandalism. The visual map helped to concentrate the project efforts on the least preferred areas and to obtain visible changes that would, at the same time, facilitate the accessibility of the green–blue area, which is a critical point according to the literature [17,21,50].

The landscape perception of the “Les Martines” community changed throughout the process triggered by the BlueHealth project intervention. The twofold strategy of improving the quality of the site through the combination of a top-down architectural intervention in the Moritz spring led by the City Council and a bottom-up process of awareness-raising, participatory decision making and hands-on actions was successful. The surveys carried out before and after the intervention among the general population of Rubí demonstrated the project’s positive impact on the community’s perception, sense of belonging and use of the space. 

### 4.1. Significance and Impact of Our Results

Our study contributes to the discussion on the importance of green–blue infrastructure and its impact on health and wellbeing in urban landscapes from the perspective of the reinterpretation of cultural landscapes related to water. The overall aim of the BlueHealth project is demonstrated in the Can Mortiz case study by the recovery of the historical memory of the site and its once significant relationship with water features. Improving health and wellbeing was at the inception of the first urban settlements in the valleys of the Vallès, Catalonia [51]. Both the purpose of those first cultural landscapes and their settlement system located along streams and springs seem to have fallen into oblivion, despite the local initiatives of mapping and research targeting the natural heritage around water and urban settlements [52]. 

The actions catalysed by the BlueHealth project with the local community and partners offered a new awareness of the site’s history and the settlement morphology to uncover the public health potential of the obsolescent, but still existing, cultural and natural landscapes. Today, the once-valued activity of socializing around the springs is in decline. The leisure opportunities offered by the springs is part of a cultural heritage that related people to nature during the previous century; it gave a purpose to walking for recreation and discovering the landscape through the act of visiting both public and private springs, as milestones in a nature itinerary—the relationship with nature experienced as a frequent social activity, including basic cooking, eating and singing. The inhabitants of many of the areas, such as the “Les Martines” valley, once famous for their lush vegetation and abundant water features, seemed to be ignorant of their origins. A loss of collective memory and physical and cultural landscape heritage was apparent [53]. 

The “urban acupuncture” and tactical urbanism actions contributed to the disparate community’s connection by caring for and constructing a new common place around the stream and the spring. Beyond the importance of the action, as triggering a social process, the Can Moritz spring masterplan aims to revive the spring as a central hub for the valley community. Water and leisure, along with green–blue infrastructure, are offered as the community’s new focus to help it to reconnect past and future through the twin threads of public health and landscape enjoyment.

Beyond the contributions to increasing the local landscape awareness arising from an integrated landscape approach recommended for implementing the ELC [54], the case study improved the qualitative perception of the intervention area by its immediate inhabitants and other people living in Rubí. The literature can demonstrate the benefits of direct contact with green and blue spaces. For instance, in a recent study including more than 18,000 participants from 18 different countries, we observed that individuals with common mental health disorders, such as depression and anxiety, were more likely to use nature for self-management [55]. In a randomized crossover study including office workers, we also observed evidence of the positive effects for wellbeing and mood of walking for 20 min alongside a blue space, in this case, the beach [56]. In yet another recent study, we showed that facilitating access to a river and improving the area translated into an overall increase in the users of an urban riverside after renovation; that the proportion of females, adults, children and ethnic minority users increased; and that locals perceived the river to be beneficial for their health and wellbeing [17]. We even estimated the derived health benefits of improving the ecological quality of a river area and facilitating access to it, and the amount of money saved by the public health system owing to this intervention [21]. 

Indeed, increasing research highlights the role of the quality of the natural environment in mental health and wellbeing [9,10,11,12,13,16]. Thus, we expect that an intervention in a local green–blue site, such as the Can Moritz spring area, will contribute to the wellbeing of the inhabitants of the “Les Martines” valley—and after more people find out about it, the level of use, overall, may increase as may, with this, the benefits. Moreover, such interventions can be instrumental for a nature-based social prescription [57] and nature-based interventions in urban contexts to improve mental health and wellbeing [58]. Besides this, the increased recognition that urban and natural environments contribute to our health and the results of the present case study support and provide the critical potential for enhancing and prioritizing community-level interventions, which are more effective than individual-level interventions [59,60,61]. The Can Moritz case study will contribute to generating and transmitting tools and knowledge so that community interventions can be implemented to improve public health by promoting and improving urban landscapes (see additional tools in https://bluehealth2020.eu/resources/toolbox/).

### 4.2. Strengths and Limitations

The Can Mortiz case study revealed the potential of an integrated sequence of methodology and techniques to strategize a low cost, tactical and participatory “urban acupuncture” intervention. Combining procedures from the landscape architecture and environmental epidemiology disciplines resulted in a sequence of actions that could easily be replicated in different urban contexts. The replicability, low cost and effectiveness of the overall method is the main strength of the project. 

The chosen Steinitz method had already been pilot tested in diverse contexts, led by two of the authors, in previous editions of the same Masters’ Workshop. This confidence in the method made it suitable to be applied as part of the case-study methodology, relying on its strength in being adapted to smaller sites, whilst knowing that the nature of the working hypothesis formulated by the experts would remain similar to that in previous case studies tested in earlier versions applied in both larger and smaller areas. 

According to both previous experiences in applying this method and the contradictory findings in the results of this case study noted earlier, we believe that the main challenge for the future is to refine the results. This further distillation could be embraced with sectoral perceptual studies, to complement the visual perception. The interpretation of the results leads to the appreciation of all the senses in perception, finally representing VPM perceptions, which are more than visual, such as pleasant or unpleasant smells and different noises and sounds. 

Furthermore, the lack of public spaces and a low socialization culture in the community limited the scope of the results obtained by the Steinitz method. The recruitment of volunteers in previous applications had been triggered mostly by casual contacts between researchers and inhabitants, which enriched the number and diversity of profiles for research. In future contexts of limited access to inhabitants, this limitation could be alleviated by social media data harvesting to balance quantitative information following international recommendations [62,63]. It would be equally interesting to mitigate the limitations in participant recruitment by fully integrating complementary perceptual studies focusing on the qualitative multisensorial experience of place proposed by Grout as teaching methods, prioritizing the role of qualitative information with few triangulating subjects [64].

Beyond the VPM, a second survey would have been useful to complete the impact evaluation cycle. However, due to limited resources, we could not conduct an observational survey to evaluate the actual use of the studied site (e.g., using the SOPARC tool [65] or others designed by the BlueHealth project [66,67]); this information would also have been valuable for estimating the levels of physical activity and understanding the new activities and social interactions occurring in the Can Moritz spring area. The postintervention survey took place a few weeks before the lockdown due to the COVID-19 pandemic in March 2020; this limited the number of people who could participate in the survey, as most did not reply before the lockdown and, therefore, their visits to the site would have been biased. Nevertheless, we are aware that the site’s use dramatically increased once people could start going out once more. 

Another limitation of the project was the low level of community engagement. Even though the efforts of dissemination and the protocols followed were correct, the low density of the area, the lack of cohesion within the community and the low degree of identification with their immediate landscape or place limited the number of participants in the surveys (in total, 86 before the intervention).

## 5. Conclusions

The Can Mortiz spring case study explored the limits of methodologies in addressing visual preference as the basis of landscape identification and tested the relationship between perception and landscape awareness-raising. We demonstrated that participatory landscape architecture planning methods with residents and other stakeholders could help to prioritize and co-design interventions. In an integrative approach, the sequence of those methods, also using tools and knowledge from health disciplines, is the main case-study contribution. Beyond these, the Can Moritz case study promoted the recovery of an ancient cultural landscape and its intangible value associated with water heritage. Promoting the health benefits to be obtained from leisure should aim for European and worldwide landscape management as part of the promotion of better health and wellbeing as identified in the Sustainable Development Goals (SDGs). 

## Figures and Tables

**Figure 1 ijerph-18-01709-f001:**
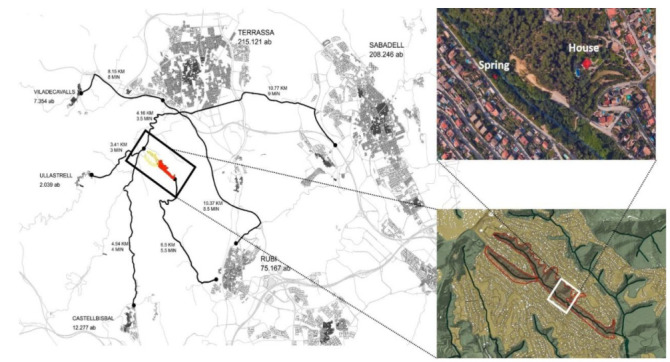
Location where the study site was conducted (valley of “Les Martines” in Rubí (Catalonia, Spain); source: Landscape Intervention and Heritage Management (LIHM) students). Yellow indicates the initiation of the stream; red indicates the area of study.

**Figure 2 ijerph-18-01709-f002:**
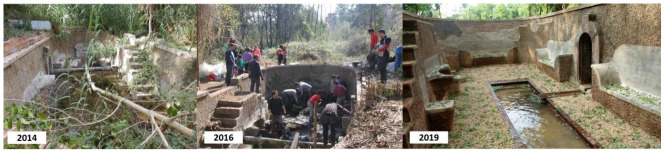
The spring of Can Moritz when first cleared of vegetation in 2014, during reconstruction in 2016 and after the restoration was completed in 2019 (pictures from Jordi Simó and Jordi Muntan).

**Figure 3 ijerph-18-01709-f003:**
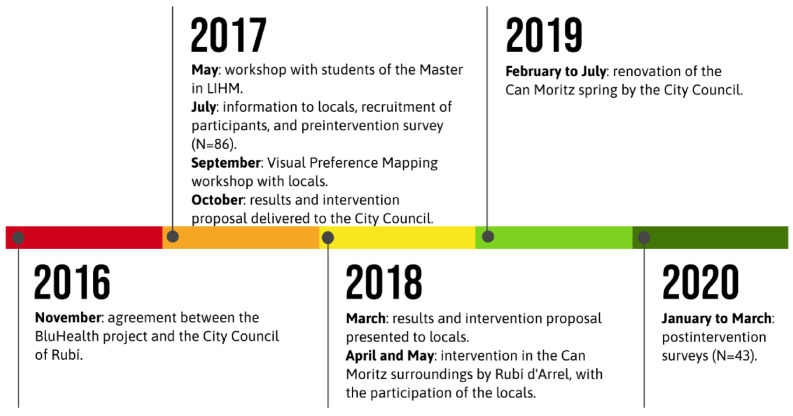
Calendar of the project (intervention and its evaluation).

**Figure 4 ijerph-18-01709-f004:**
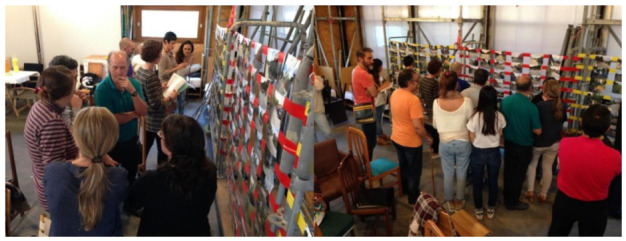
Participatory workshop with the neighbours (September 2017)**.**

**Figure 5 ijerph-18-01709-f005:**
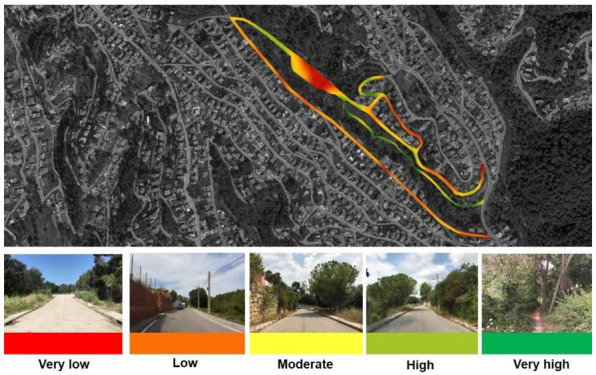
The visual preference mapping generated during the workshop with the locals (colour gradient from least to most preferred areas: red indicates areas of improvement, and green indicates areas with potential).

**Figure 6 ijerph-18-01709-f006:**
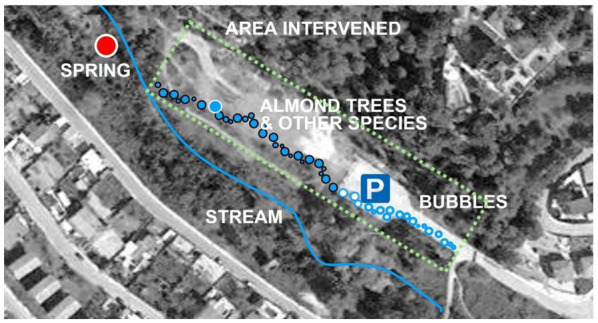
Masterplan of the interventions in the spring’s surroundings following the conclusion of the visual preference mapping (VPM) analysis (P indicates parking lot area). The bubbles are blue circles painted onto the surface of the car park, and the other circles are the positions of the painted logs.

**Figure 7 ijerph-18-01709-f007:**
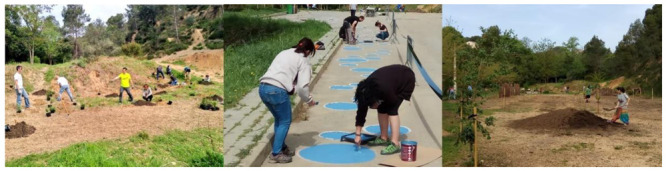
Intervention in the surroundings of the spring, with the participation of the locals (clearance of rubbish, painting of circles on the car park surface, and tree and shrub planting).

**Table 1 ijerph-18-01709-t001:** Descriptive table of the survey participants.

	Preintervention (All Participants, N = 86)	Preintervention (Only Including Those Also with Postintervention Information, N = 43) ^1^	Postintervention (N = 43)
**Age (mean (min–max))**	46.8 (21–77)	43.7 (22–74)	NV
**Female (%)**	45.4	44.2	NV
**Self-considered from a minor ethnicity (%)**	8.1	2.3	NV
**Married or living in couple (%)**	80.2	81.4	62.8
**University degree (%)**	47.7	51.2	NV
**Working (%)**	66.3	79.1	74.4
**Work in Rubí (%)**	43.9 (out of N = 57 working)	44.1 (out of N = 34 working)	28.1 (out of the N = 32 working)
**Retired (%)**	15.1	4.7	7.0
**Access to a garden at home (community or private garden, %)**	94	93	87.3
**General health (%)**			
Very good	24.4	25.6	25.6
Good	43.0	48.8	46.5
Normal	24.4	20.1	25.6
Bad	7.0	4.7	2.3
Very bad	1.2	0.0	0.0
**Participate in a local NGO, organization, assembly or entity (%)**	33.7	51.2	44.2
**Have a dog (%)**	48.8	37.2	37.2
**In the last 12 months, how often, on average, have you spent your free time in green and blue spaces? (%)**			
Everyday	15.1	16.3	11.6
Several times a week	36.1	44.2	30.2
Once a week	11.6	14.0	21.0
Once or twice a month	19.8	16.3	23.3
Several times in the last 12 m	15.1	7.0	14.0
Never in the last 12 m	2.3	2.3	0.0
**Know the Can Moritz spring in Rubí (%)**	64.0	60.5	83.7
**Quality of the Can Moritz spring (%)**	**N = 55**	**N = 26**	**N = 36**
Very good	0.0	0.0	13.9
Good	20.0	26.9	38.9
Not good, not bad	34.6	34.6	30.6
Bad	34.6	30.8	16.7
Very bad	10.9	7.7	0
**In the last six months, how many times have you visited the spring? (%)**	**N = 55**	**N = 26**	**N = 36**
Never	34.6	30.1	30.6
Once or twice	45.5	50.0	52.8
Between 3 and 6 times	12.7	11.5	8.3
Seven or more times	7.3	7.7	8.3
**In the last four weeks, how many times have you visited the spring? (%)**	**N = 36**	**N = 18**	**N = 25**
Never	61.1	66.7	68.0
Once or twice	33.3	22.2	24.0
Once a week	5.6	11.1	4.0
Several times a week	0.0	0.0	4.0
	**N = 14**	**N = 6**	**N = 8**
**Date of the last visit**	Jan, Feb, Mar, June, Sept, Dec	Feb, Mar, June, Sept	Jan, Feb, Mar ^a^
**Time spent (mean (min–max))**	20.4 min (5–90 min)	15.8 min (5–30 min)	32.5 min (10–60 min)
**Activities (%)**	**N = 14**	**N = 6**	**N = 8**
Bike	7.1	0.0	0.0
Running	28.6	33.3	0.0
Nordic walking	7.1	0.0	12.5
Observing fauna	7.1	16.7	12.5
Walking with the dog	28.6	16.7	37.5
Walking with a dog	21.4	33.3	25.0
Eating or drinking	0.0	0.0	12.5
**Number of adults (>16 years, %)**	**N = 14** **** **(35.7% with <16)**	**N = 6** **** **(50% with <16)**	**N = 8** **** **(50% with <16)**
One	57.1	50.0	25.0
Two	35.7	50.0	62.5
More than two	7.1	0.0	12.5

NV: non-variable throughout time; <16: children under 16 years of age.^a^ We had to stop the postintervention evaluation due to full COVID-19 lockdown; people were not allowed to be outdoors.

**Table 2 ijerph-18-01709-t002:** How survey respondents felt about their visits to the Can Moritz spring (it only includes respondents who reported visiting the spring in the last four weeks).

(%)	Totally Agree	Agree	Agree a Little Bit	Neutral	Disagree a Little Bit	Disagree	Totally Disagree
**Initial study population (with preintervention information)**							
**Preintervention (N=14)**							
**I felt satisfied with the visit**	4.1	14.3	14.3	14.3	14.3	21.4	14.3
**I felt part of nature**	14.3	14.3	7.1	14.3	14.3	28.6	7.1
**I felt safe (i.e., I felt protected)**	21.4	14.3	7.1	35.7	0.0	7.1	14.3
**The area was free of rubbish and vandalism**	7.1	0.0	0.0	7.1	42.9	14.3	28.6
**It had good facilities (e.g., parking, roads, bathrooms, fountains drinking water, barbecues)**	0.0	0.0	7.1	7.1	7.1	42.9	35.7
**Study population with pre- and postintervention information**							
**PREINTERVENTION (N=6)**							
**I felt satisfied with the visit**	16.7	33.3	0.0	16.7	0.0	16.7	16.7
**I felt part of the nature**	33.3	33.3	16.7	0.0	0.0	16.7	0.0
**I felt safe (i.e., I felt protected)**	50.0	16.7	0.0	0.0	0.0	16.7	16.7
**The area was free of rubbish and vandalism**	16.7	0.0	0.0	0.0	33.3	16.7	33.3
**It had good facilities (e.g., parking, roads, bathrooms, fountains drinking water, barbecues)**	0.0	0.0	0.0	16.7	0.0	33.3	50.0
**POSTINTERVENTION (N=8)**							
**I felt satisfied with the visit**	62.5	12.5	12.5	12.5	0.0	0.0	0.0
**I felt part of the nature**	50.0	37.5	125	0.0	0.0	0.0	0.0
**I felt safe (i.e., I felt protected)**	37.5	37.5	0.0	0.0	25.0	0.0	0.0
**The area was free of rubbish and vandalism**	12.5	37.5	25.0	0.0	0.0	0.0	25.0
**It had good facilities (e.g., parking, roads, bathrooms, fountains drinking water, barbecues)**	0.0	25.0	0.0	12.5	12.5	25.0	25.0

## Data Availability

The data presented in this study are available on request from the corresponding author. The data are not publicly available due to restrictions, e.g., privacy or ethical.

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
