# Peer review of "A Transdisciplinary Approach to Recovering Natural and Cultural Landscape and Place Identification: A Case Study of Can Moritz Spring (Rubí, Spain)"

_ijerph, 2021, doi:10.3390/ijerph18041709_

Round 1

Reviewer 1 Report

I enjoyed reading this paper, primarily because it is direct, focusses on the issues, and draws sensible conclusions. I thought it could probably be shortened in places - the introduction and discussion sections are longer than they really need to be for a case study. It was useful to have the detail in the tables, although they were a bit hard to follow. Some more work on the formatting might help (use lines, in Table 1, as in Table 2), and I wondered if the small sample sizes in places meant that it would be better to simply give the actual counts rather than the %s. There should be a solution to the repetition in the statements at left. 

First paragraph of results appears to be author instructions from the journal. Steinitz is a reference and does not need the "Professor Carl".

There was some conversational English in places that could be shortened or eliminated. For example, the first paragraph of Section 2.2 is probably entirely unnecessary. Historical detail of the progress of the study creates time wasting for the reader. I see that there seem to be some first language English speakers in the author list, so they should be able to help with that. 

Figure 3 could be incorporated into text and would take less space. 

Captions need to be a little more detailed in order to properly inform the reader about the content of the table or figure. There is no caption for the photos at line 340. 

Reviewer 2 Report

The article presents a valuable case study of research on the perception of the landscape. The first main concern of the Reviewer is that it does not fit the journal. The relationships with the theme of urban health are very scarce and while this topic is mentioned in the introduction and in the discussion (as references to other research by the same authors), the actual research is on something else. This discrepancy is the primary issue and in my opinion, the article should be shifted to some other journal which addresses the perception of the landscape. The overlap of two (or more) topics makes the article lack focus. Moreover, partially due to the above problem, the research background is not properly presented. The results are not properly discussed - instead, the Authors introduce the framework of urban health which has not been researched.

Taking the above into consideration, the Reviewer proposes rejection of the current submission. 

Reviewer 3 Report

Dear authors

Many thanks for this article. It is good to see more qualitative research in landscape architecture addressing health and wellbeing aspects, particularly related to cultural and social aspects. I think this is a great piece of research, therefore, my comments aim at strengthening the entire argument. I believe the introduction as well as some aspects of the results and discussion should be addressed to greater clarity. A detailed overview is followed:

Abstract:

Well written and provides a great summary of the research undertaken. The transition to the last sentence in the abstract is a bit odd. I’d encourage the authors to check L36-40.

Introduction

Well written and structured. The first two paragraphs are very informative but lack to cite very important research done around health and wellbeing. For instance, green spaces directly affect our own health and wellbeing, but many studies show that cultural and social aspects contribute to our physical and psychological health. To name a few of more qualitative research that I’d urge the authors to include:

Andrews, G. (2017). Landscapes of Wellbeing. In T. Brown, G. J. Andrews, S. Cummins, B. Greenhough, D. Lewis, & A. Power (Eds.), Health geographies: a critical introduction (pp. 59-74). John Wiley & Sons.

Bell, S. L., Foley, R., Houghton, F., Maddrell, A., & Williams, A. M. (2018). From therapeutic landscapes to healthy spaces, places and practices: A scoping review. Social science & medicine, 196, 123-130.

Foley, R., & Kistemann, T. (2015). Blue space geographies: Enabling health in place. Health & Place, 35, 157-165.

Kershaw, C., McIntosh, J., Marques, B., Cornwall, J., Stoner, L., & Wood, P. (2017). A potential role for outdoor, interactive spaces as a healthcare intervention for older persons.

Marques, B., McIntosh, J., & Chanse, V. (2020). Improving Community Health and Wellbeing Through Multi-Functional Green Infrastructure in Cities Undergoing Densification. Acta Horticulturae et Regiotecturae, 23(2), 101-107.

Marques, B., Freeman, C., Carter, L., & Pedersen Zari, M. (2020). Sense of Place and Belonging in Developing Culturally Appropriate Therapeutic Environments: A Review. Societies, 10(4), 83.

Souter-Brown, G., Hinckson, E., & Duncan, S. (2021). Effects of a sensory garden on workplace wellbeing: A randomised control trial. Landscape and Urban Planning, 207, 103997.

Thompson, C. W. (2011). Linking landscape and health: The recurring theme. Landscape and urban planning, 99(3-4), 187-195.

Ward Thompson, C., Aspinall, P., & Bell, S. (2010). Innovative approaches to researching landscape and health. Open Space: People Space II. New York.

Some of these works will permit you to make a stronger link with the values highlighted through the ELC. This link also needs to be further expanded. Why is the ELC so important? What is the contribution of this convention to the study? Why aren’t other international policies discussed/adopted? The WHO has many documents that may be of importance.

Materials and Methods

This section is clear and provides a good overview of the case-study. Section 2.1 currently misses supportive references.

Results

I believe the first 3 lines are from the provided journal template. If this is the case, the authors need to introduce the results – what are the overarching themes that the reader should be expecting?

Section 3.1 is well explained as well as section 3.2. For the table provided in L350, some of the questions should be simplified for easy-read of the table.

Discussion

The authors were able to draw the most important conclusions of this study. This section is clear and well-articulated. I’d like the authors to expand the section pertaining to strengths and limitations, for instance, the method introduced by Steinitz has suffered changes and more robust processes are now in place. Did the authors find the ‘Steinitz method’ to be a limitation? What are the key aspects that need to be change in light of the case-study? What other aspects in this research could have potential for further development and improvement? Just some ideas.

Conclusion

Short and concise.

Reviewer 4 Report

The authors investigate a transdiciplinary approach to recover natural and cultural landascape and place identification ,focusing on the case study of Can Moritz spring.

The proposed study is interesting but there are some points that the authors should better discuss.

The authors should be better described the novelties of their study with respect to existing ones. In particular, the author should discuss limitation and cons of the examined approaches. Furthermore, the authors should provide more details and discussion about the obtained results. The Discussion section also needs to be improved by analyzing the outcome of evaluation section.

I suggest to further analyze more recent approaches about the examined topics. In particular, I suggest the following papers to investigate multimedia content for cultural heritage:

1) Kira: a system for knowledge-based access to multimedia art collections. In 2017 IEEE 11th international conference on semantic computing (ICSC) (pp. 338-343). IEEE.

2) Recommendation in social media networks. In 2017 IEEE Third International Conference on Multimedia Big Data (BigMM) (pp. 213-216). IEEE.

Finally, I suggest to perform a linguistic revision.

Reviewer 5 Report

In this article, the authors provide an interesting and methodologically significant analysis of their heritage management/revitalization case study, Can Mortiz. The transdisciplinary approach they follow involves a range of approaches including traditional landscape analysis, geography, landscape architecture, and historic preservation techniques coupled with a grounded theory through their onsite mapping and qualitative (stake holder involvement and survey) analysis. The background literature and theoretical review provide a good methodological context and the application of these techniques are well written and understandable.

It is a well-structured analysis and the results and conclusions are clear. There are a few minor issues and/or changes that I see as important to strengthen this article. The following is a line by line identification of issues.

In line 26 ff, there is a critique of the lack of training for students in Landscape Architecture related to the issues of place and the cultural landscape. This is a critique I heard 20 years ago as well. This critique in 2021 either means that the discipline has not changed (which is a negative for the field), or this is an outdated trope-ish critique that might turn off readers in Landscape Architecture. One of the target audiences of this article seems to be Landscape Architects, so a verification of the critique seems warranted.

Line 63: the word exponentially seems to not really fit here without further explanation.

Line 79: A work that should be included and discussed in this article is Roe and Taylor 2014 New Cultural Landscapes (Routledge). A number of chapters are directly relevant to this study, especially Roe’s focus on ordinary (what she calls “everyday”) landscapes

Line 80 ff: the example given is from US, but the preceding paragraphs are dedicated to the EU. Is there an early European example of conservation/preservation that can be used in tandem with the US example to align it with the ELC discussion?

Line 163: There needs to be a deeper discussion into the origins and applications of previous studies of “urban acupuncture.”

Line 222: I do not know if it needs to be addressed, but this process is built upon the foundations of Carl Sauer’s Cultural Landscape (1925) approach.

Line 428. The placement of Table 2 interrupts the flow of the text too much. I feel it could be more strategically placed for readability.

Line 522 ff needs revision. “Studied” is floating and confusing. In general, the entire significance and limitations section should be revised for issues of clarity.

Round 2

Reviewer 4 Report

I think that the authors have addressed all my concerns